# Applying Quantum Mechanics for Extreme Value Prediction of VaR and ES in the ASEAN Stock Exchange †

**Chukiat Chaiboonsri *** and **Satawat Wannapan**

Modern Quantitative Economic Research Center (MQERC), Faculty of Economics, Chiang Mai University, Chiang Mai 50200, Thailand; lionz1988@gmail.com
* Correspondence: chukiat1973@gmail.com
† This research work was partially supported by Chiang Mai University.

**Abstract:** The advantage of quantum mechanics to shift up the ability to econometrically understand extreme tail losses in financial data has become more desirable, especially in cases of Value at Risk (VaR) and Expected Shortfall (ES) predictions. Behind the non-novel quantum mechanism, it does interestingly connect with the distributional signals of humans' brainstorms. The highlighted purpose of this article is to devise a quantum-wave distribution methodically to analyze better risks and returns for stock markets in The Association of Southeast Asian Nations (ASEAN) countries, including Thailand (SET), Singapore (STI), Malaysia (FTSE), Philippines (PSEI), and Indonesia (PCI). Data samples were observed as quarterly trends between 1994 and 2019. Bayesian statistics and simulations were applied to present estimations' outputs. Empirically, quantum distributions are remarkable for providing "real distributions", which computationally conform to Bayesian inferences and crucially contribute to the higher level of extreme data analyses in financial economics.

**Keywords:** quantum mechanics; wave function; extreme value analysis; Bayesian inference; stock market; Value at Risk (VaR); Expected Shortfall (ES); prediction



## 1. Introduction

Physicists' interest in the social sciences is not novel. The word "econophysics" is the perspective applied to economic computational models and concepts associated with the "physics" of systematical complexity—e.g., statistical mechanics (quantum mechanics), self-organized criticality, microsimulation, etc. (Hooker 2011). Fundamentally, most econophysicists have in mind that the approach—computational physics for econometrics—seeks to structure physically realistic models and theories of economic phenomena from the actually observed features of economic systems. Practically, econophysicists and economists are connected by analyzing financial markets, but the problem is that they are trained in "different schools" (Ausloos et al. 2016). The exploration of the tangible of computational physics and financial econometrics is the nexus of this paper.

Not surprisingly, physics students have been trained and have known that the frontier of modern physics uses plain language—quantum physics and relativity (Bowles and Carlin 2020). The power of quantum physics substantially existed in the chaotic period of World War II—reasonably called the "beginning" of modern physics—by Werner Heisenberg, who was a founder of quantum mechanics and a significant contributor to the physics of fluids and elementary particles (Saperstein 2010). With this great exploration, modern quantum physics has brought people to distinctly shift the standard of living through numerous inventions such as microwaves, fiber optic telecommunications, super computers, etc.

In economics, "Marshall plus Keynes" neoclassical synthesis is still teachable for the non-specialist future citizen since the "visible hands" stated by Adam Smith has been elusive, and the story has therefore never been all that easy to see—as a perusal of his original text demonstrations (Persky 1989). However, human decision-making processes

are significantly from their "power" explained by integrating psychological aspects and individual social-economic ideas (Sijabat 2018). From this perspective, quantum physics can potentially affect a merger with computational economic models through the concept of the power behind a decision. More expressly, Figure 1 displays the diverse iceberg for economic movements. Along with fluctuations in the trend, traditionally computational economics restrictedly captures only the observable zone (the top of the iceberg). However, numerous amounts of information exist underneath the water, which potentially motivate human perspectives to conclude a final decision, are intentionally neglected by the assumption called "normality". With this strongly theoretical supposition, traditional econometrics has been trustworthy for more than a hundred years.

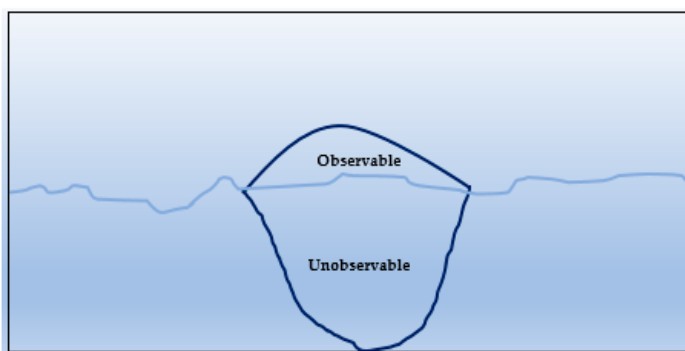

**Figure 1.** Diverse economies iceberg.

Unfortunately, there have been many economic collapses after the industrial revolution in the mid-18th century. Economists and econometricians are blemished with their predictive foresight's computational failures, especially forecasting in financial investments. The deep root of the problem is about their fundamental thoughts. Thinking as a traditional econometrician is to model observed information by a random walks model, the easiest way to imitate rational human aspects. Box 1 represented in Figure 2 displays the concept that the random walk model is the logic to reach B from A. This principle's systematic idea is to sample only a static spot when the arrow is tangible B. However, the critical query is that this fundamental cannot seemingly support the existence of human thinking.

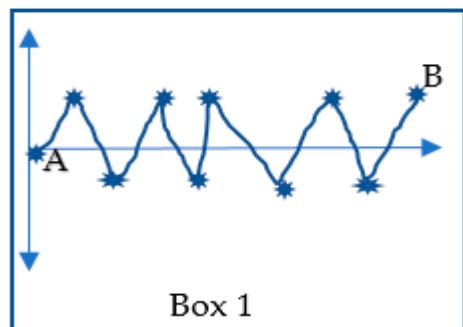 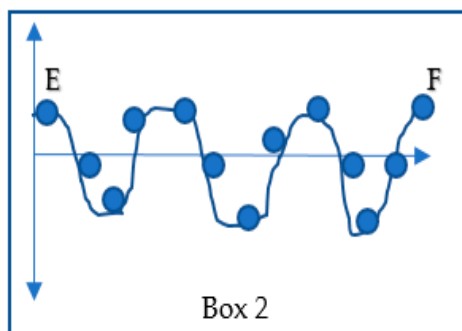

**Figure 2.** Two different concepts between Random walks and wave function (Quantum walks).

Human decisions are sourced from electromagnetic waves and particles since cell-to-cell communications occur through a process known as "synaptic transmission", where chemical signals are passed between cells generating electrical spikes in the receiving cell. To think like modern econophysicists (Quantum Mechanics: Wave-particle duality), it is reasonable to state the arrow from E to F displayed in box 2 is not a random walk model (Quantum walk process). The observation at the point F exists when it passes the process of dynamic wave movements. In other words, the reason to decide to select F is the origin of numerous latencies, which are attitudes, perspectives, morals, etc. Interestingly, this

fundamental can potentially raise a novel horizon for observing information using modern econometric (Quantum econometrics) estimations and predictions.

Financial markets have several of the properties that characterize complex systems and interact nonlinearly in the presence of estimations (Mantegna and Stanley 2000). One of the interesting areas in finance is the pricing of derivative instruments. The graphical trend displayed in Figure 3 shows the example of dynamic stock exchanges in Thailand between 2000 and 2020—investors have a variety of reasons and decide the process of brainstorms. The graph implies that it is identical to a wave transmission. Hence, it is time to seek an alternative tool to compute this kind of complex data econometrically.

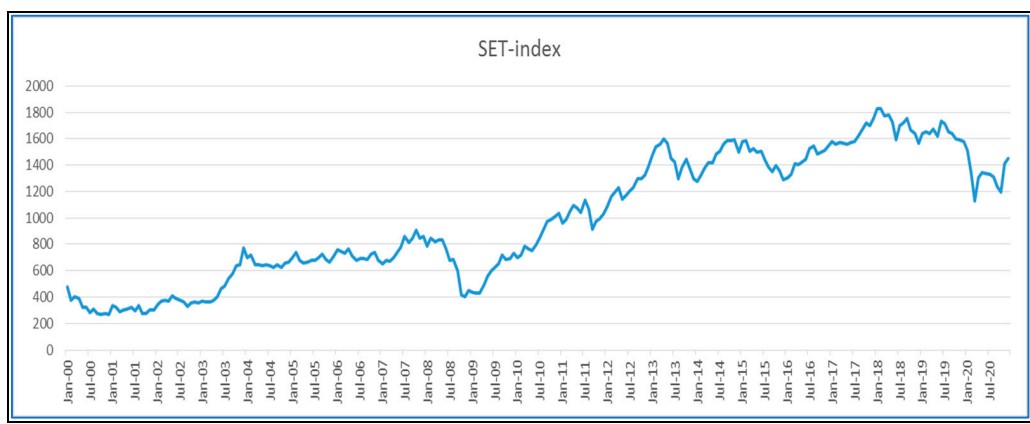

**Figure 3.** The dynamic movement of the Thailand stock exchange between 2000 and 2020. Source: Stock Exchange of Thailand (SET)

This research tries to fill the research gaps between the traditional econometrics method and modern econophysicists (Quantum Econometrics), which apply in financial markets, especially the extreme value prediction in the ASEAN stock exchange. However, this research is organized, as follows, by explaining the conceptual framework between traditional econometrics and Quantum Econometrics (Modern Econophysicist). The second part is how to apply this conceptual framework in extreme value prediction, especially the extreme value of Value at Risk (VaR) and Expected Shortfall (ES) of five stock markets in ASEAN countries. The last part of this research is an exclusive summary for comparison between two methods to forecast the extreme value of Value at Risk (VaR) and Expected Shortfall (ES) in five stock markets of ASEAN countries based on Risk management analysis.

## 2. Literature and Critical Thinking

It is not simple to exactly explain and picture humans' decisive believes or faiths. In terms of a mathematical expression, deductive logic was the ideal invention trying to reach a conclusion. However, thoughts are not logically controllable as similar to a robotic mechanism. In econometrics, this is defined as "extreme distribution". To make sense of it, the root of distributional generating is interesting to reconsider.

### 2.1. The Origin of Quantum Distributions

Inside of the area of traditional econometrics, the original process of a random walk based on the scaling limit, which generalizes the so-called iterated Brownian motion, is useable and acceptable academically. This theoretical concept of the random walk graphically displayed in Figure 4 was considerably generalized and extended by the Polish physicist Marjan Smoluchowsk (Kac 1947). It is continuously modified to be functional in modern quantitative research. Jung and Markowsky (Jung and Markowsky 2013) showed the random walk's advantage at random times to be considered the "alternating random", which rewards the schematic scale to indicate fractional stable motions. Although the theory of random-walk processes is continuously acceptable, the theory has started to be

criticized. The weakness of random walk algorithms is stated by (Saghiri et al. 2019). The non-intelligent random walk models may not be a problem-solving method in real-world problems since some complex systems such as biological networks or social networks work as a "learning mechanism". It seems the performance of random-walk processes is low when used to explain mechanical information about the practical problems' nature.

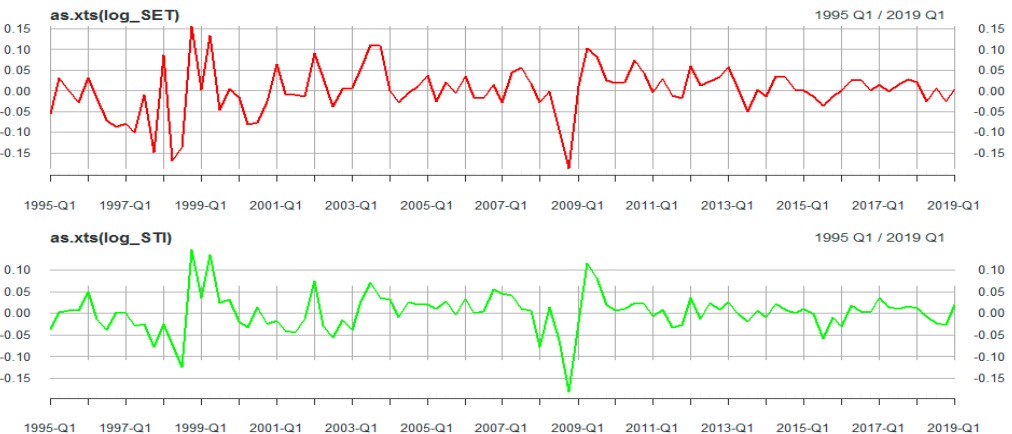

**Figure 4.** The example of random walk patterns based on Brownian motion. The data visualize two log-return sets of quarterly stock indexes in Thailand (SET) and Singapore (STI) between 1995Q1 and 2019Q1. The fluctuation of index trends is the difference in pitch between positive and negative values, but it inclines stationary (close to 0 and unchangeable) in the long-term consideration.

Modernly, there are many attempts to make a reconsideration in the impetus of humans' decision calls. As Adam Smith said, invisible hands are behind the scenes. This undetectable power is deliberately linked to quantum behaviors. Quantum mechanics based on Newton's motion laws are good enough to predict how behavioral complexities are inspired. Newton's laws are seen to be consequences of the fundamental way the quantum world works (Ogborn and Taylor 2005). However, in accounting for small occurrences, Newtonian physics' failure is evident at the atomic level. This implies that the lack of precise calculations cannot be simply captured by Newtonian quantum computing. It turns out that the "Hamiltonian formulation" is the formalism that most readily generalizes to quantum mechanics via the Schrodinger equation (Piziak and Mitchell 2001), which is the crucial fundament for exploring quantum key distributions

One of the highlighted obstacles in quantum key distributions is capturing and pointing signals of quantum mechanism exactly. As stated in the contribution by (Bruß and Lütkenhaus 2000), the problem of cross-polarized cryptography between the two polarization modes and a random (classical) rotation of the polarization along the propagation direction is informationally detected by using Ekert's privacy amplification (Ekert et al. 1994). Interestingly, the quantum distribution is being a truly evolutional data analysis for post-modern econometrics.

### 2.2. Quantum Computing in Financial Econometrics

The Hamiltonian formulation for the time-independent Schrödinger equation composes a quantum evolutional enlargement of the classical harmonic oscillator approaches to economics' business cycle dynamics. As the literature on Piotrowski and Słodkowski (2001); Choustova (2007); Gonçalves and Gonçalves (2008); Choustova (2009); Gonçalves (2013); and Gonçalves (2015); Chaiboonsri and Wannapan (2021), a quantum application to extreme financial optimizations, therefore, contributes to the novel discussion within forecasting financial economics and raises a criticism to the empirical validity of the geometric Brownian motion and geometric random walk models of price dynamics, which is commonly employed in financial economics as mathematical tools for solving pricing problems, especially risks and returns analyses.

## 3. The Objective and Scope of Research

The risk and return of financial markets are the main investigations of the paper. Quarterly data from 1995 to 2019 were observed. Five major stock exchanges in five ASEAN countries such as Thailand (SET), Singapore (STI), Malaysia (FTSE), Philippines (PSEI), and Indonesia (PCI) were processed in three sections of the research framework, including data visualization, risks management (Value at Risk: VaR), and returns forecasting (Expected Shortfall: ES). Technically, parametric estimations' main statistical tool is the subjective method called "Bayesian inference", and observations are deliberately focused on the extreme tail loss of distributional portraits, theoretically known as "extreme value".

The scope of the research processes is displayed in Figure 5. Expressly, the observations are processed to the section of data visualization (screening data). Descriptive explanation, stationary testing, and normality checking are the main consideration, and then the raw data is modified by two critical concepts—a random walk (Gaussian) distributional set and quantum-wave distribution. The next step is to insert two distinguished data into the function of the Generalized Pareto Distribution (GPD) extreme value analysis. Heavy loss tails are clarified and analyzed by setting the prior density for parameters at the Bayesian estimation threshold. The most precise prediction between two distributions is validated by computing the Deviation Information Criteria (DIC).

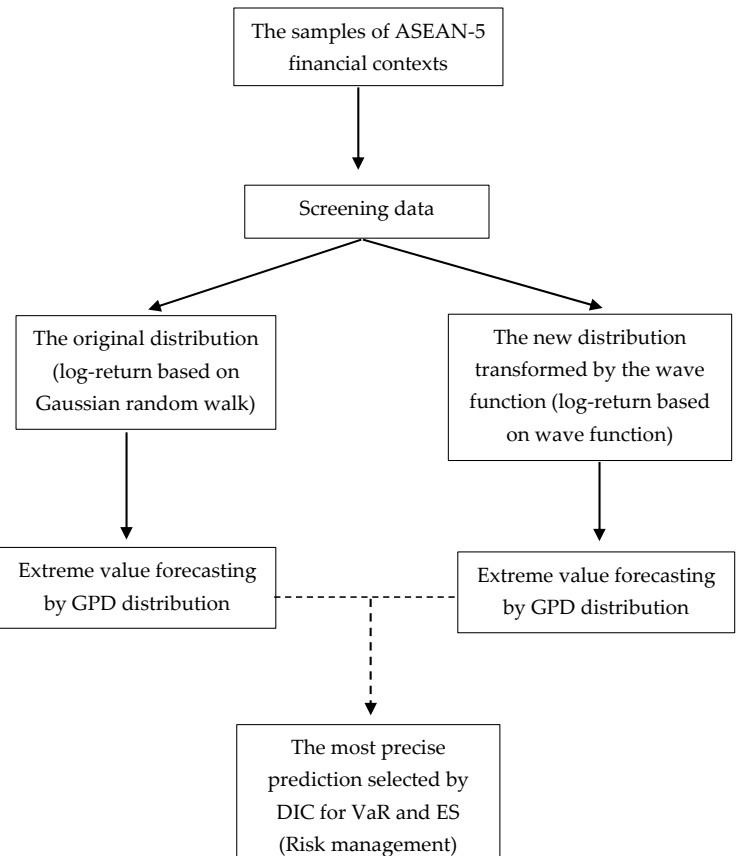

**Figure 5.** The scope of research.

## 4. Methodology

### 4.1. Quantum Mechanics and Wave Function for Time Series Movement

Since 1926, Erwin Schrödinger developed the wave function implemented to predict the quantum system's behavior or quantum mechanics, especially for the prediction of the momentum of energetic electrons (Schrödinger 1926). This quantum mechanics based on the idea of Louis de Broglie's wave–particle duality in 1924 was played an important role that has significantly influenced the development of Schrödinger's wave

function ([Recherches sur la théorie des quanta  Researches on the quantum theory](#)). Basically, energy can be expressed by a simple equation, as shown below:

$$E = KE + PE \tag{1}$$

where $E$ = Energy, $KE$ = Kinetic Energy, $PE$ = Potential Energy, $KE = \frac{1}{2}MV^2$, $M$ = Mass (kg), $V$ = Velocity (m/s). Consequently, we obtain

$$E = \frac{1}{2}MV^2 + U, PE = U \tag{2}$$

The momentum of $P = MV$, the object is empowered to move by $P^2 = M^2V^2$ then $V^2 = \frac{P^2}{M^2}$. However, Equation (2) can be rewritten by substitution of $V^2$ and this is presented in Equations (3) and (4) as follows:

$$E = \frac{1}{2}M\frac{P^2}{M^2} + U, \tag{3}$$

$$E = \frac{P^2}{2M} + U. \tag{4}$$

Once again, the wave function from Schrödinger's equation's original idea used $\psi$ to represent the energy of the particle-wave duality movement. This can be demonstrated as follows:

$$\psi = e^{i(kx-\omega t)} \tag{5}$$

$$\frac{d\psi}{dx} = ike^{i(kx-\omega t)} = ik\psi \tag{6}$$

$$\frac{d^2\psi}{dx^2} = (ik)^2 e^{i(kx-\omega t)} \tag{7}$$

$$\frac{d^2\psi}{dx^2} = i^2 k^2 e^{i(kx-\omega t)} \tag{8}$$

$$\frac{d^2\psi}{dx^2} = i^2 k^2 e^{-i(\omega t-kx)} \tag{9}$$

$$\frac{d^2\psi}{dx^2} = -k^2 e^{-i(\omega t-kx)} \tag{10}$$

and Equation (10) is defined that: $k = \frac{P}{\hbar}$, (de Broglie relation) then

$$\frac{d^2\psi}{dx^2} = -(\frac{P^2}{\hbar^2})\psi, \psi = e^{-i(\omega t-kx)}. \tag{11}$$

Equation (11) is multiplied by $\hbar^2$ on both sides, we obtain

$$-\hbar^2\frac{d^2\psi}{dx^2} = P^2\psi \tag{12}$$

From Equation (4), it can be modified, and it can be rewritten by the equation as displayed below:

$$E = \frac{P^2}{2M} + U, \tag{13}$$

$$E\psi = \frac{P^2\psi}{2M} + U\psi, \tag{14}$$

$$E\psi = \frac{-\hbar^2}{2M}\frac{d^2\psi}{dx^2} + U\psi, -\hbar^2\frac{d^2\psi}{dx^2} = P^2\psi, \tag{15}$$

Equation (15) was mentioned by Schrödinger to implement the quantum mechanics prediction for electrons moved by energy relied on the case of time-independence. For making a sensible computation, the time-dependent case can be done by starting from the Planck–Einstein relation (Griffiths 1995) as presented that $E = \hbar\omega = hf$, $E = hv = hf$, $h = $ Planck constant ($6.626 \times 10^{-34}$). The Planck–Einstein relation suggests that whenever energy is empowered, frequencies ($v$) are parallel increments, the Planck constant $h$ is stable. The proof of the following equations can express electrons' energic movements:

$$\frac{d\psi}{dt} = -i\omega e^{i(kx-\omega t)} \tag{16}$$

$$\frac{d\psi}{dt} = -i\omega\psi, \psi = e^{i(kx-\omega t)} \tag{17}$$

From the Planck–Einstein relation, we obtain

$$E = \hbar\omega, \tag{18}$$

$$E\psi = \hbar\omega\psi \tag{19}$$

Multiplied $\frac{-i}{\hbar}$ into Equation (19) on both sides,

$$\frac{-i}{\hbar}E\psi = -i\omega\psi, (\frac{d\psi}{dt} = -i\omega\psi), \tag{20}$$

$$\frac{-i}{\hbar}E\psi = \frac{d\psi}{dt} \tag{21}$$

$$E\psi = \frac{\hbar}{-i}\frac{d\psi}{dt}. \tag{22}$$

The time-independence should be transformed into the Schrödinger equation for the time-dependent by replacing Equation (22) into (15). The finalized result of these steps is Equation (23):

$$\frac{\hbar}{-i}\frac{d\psi}{dt} = \frac{-\hbar^2}{2M}\frac{d^2\psi}{dx^2} + U\psi \tag{23}$$

and

$$i\hbar\frac{d\psi}{dt} = \frac{-\hbar^2}{2M}\frac{d^2\psi}{dx^2} + U\psi \tag{24}$$

The finalized equation is the fundamental of the Schrödinger equation for predicting the momentum of wave-particle dualities in different cases. However, the right-hand and left-hand sides of those equations can be substituted by $\hat{H}$ (Hamiltonian OPERATOR, $i\hbar\frac{d}{dt}|\psi(t)\rangle = \hat{H}|\psi(t)\rangle$ (Time-dependent), $\hat{H}|\psi\rangle = E|\psi\rangle$ (Time-independent)) for forecasting the total systematic energy. In particular, the behavior for the quantum mechanism. In terms of the Schrödinger wave function's interpretation, this is the highlight for this article to apply the periodic function for measuring the momentum of returns of stock markets in ASEAN countries. Mathematically, we start with

$$\psi = A\sin(\frac{2\pi}{\lambda}x), \tag{25}$$

where $\psi$ represents the prediction value of total energy for the momentum of return movements during observable periods. $A$ is the amplitude of Equation (25) and $\lambda$ is the wavelength included in the equation simultaneously.

Figure 6 implies the concept of quantum mechanics applying for stock return predictions. In other words, this cognition is applied from the concept of sound amplification mathematically explained in Equation (25). The upper regime (high energy zone) is a quadrant of positive positions, which explains that returns are still moved. In this case, the high energy zone stands for the explanation of Bull market momentums (Ahn et al.

2018; Ataullah et al. 2008). Conversely, the fall of returns (Bear market momentum) is the negative quadrant—compared with the case of low energy with no evidence or information to push up. Interestingly, this is a huge challenge from quantum mechanics' performance to figure out the better frontier for understanding stock return fluctuations, especially when extremes data are intensively mentioned, but distribution is elusive to find.

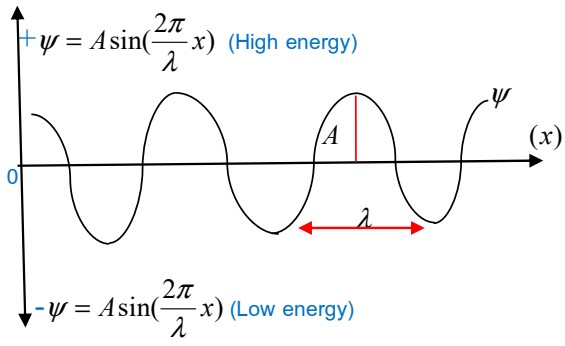

**Figure 6.** The wave function is based on the Schrödinger equation.

*4.2. Extreme Value Analysis*

Many distributions have been mentioned to model share returns as a whole observation (normal tail distributions). However, the weakness of the whole distribution is the missing of extreme tail losses. The Generalized Pareto Distribution (GDP) introduced by Pickands (1975) intentionally focuses on the threshold of the extreme losses by taking the negative of the log-returns and then choosing a positive threshold. The model assumes observations under the threshold, $\mu$, is from a certain distribution with parameters $\eta$. $H(\cdot|\eta)$ is from a GPD. Thus, the distribution function $f$ of any sample $x$ can be expressed following Behrens et al. (2004) as

$$f(x|\eta, \xi, \sigma, \mu) = \begin{cases} H(x|\eta), & x < \mu \\ H(x|\eta) + (1 - H(\mu|\eta))G(x|\xi, \sigma, \mu), & x \geq \mu. \end{cases} \quad (26)$$

For an observation of size $\eta$, $x = (x_1, ...., x_n)$ from $f$, parameter vector $\theta = (\eta, \sigma, \xi, \mu)$, $N = [i : x_i < \mu]$, and $P = [i : x_i \geq \mu]$, the likelihood equation is

$$L(\theta; x) = \prod_N H(x|\eta) \prod_P (1 - H(x|\eta)) \left[ \frac{1}{\sigma} \left( 1 + \frac{\xi(x_i - \mu)}{\sigma} \right)_+^{-(1+\xi)/\xi} \right], \quad (27)$$

for $\xi \neq 0$, and for $\xi = 0$,

$$L(\theta; x) = \prod_N H(x|\eta) \prod_P (1 - H(x|\eta))[(1/\sigma) \exp\{(x_i - \mu)/\sigma\}].$$

The threshold $\mu$ is the point where the density has a disruption. Depending on the parameters, the density jump can fluctuate positively or negatively, and in each case, the choice of which observations will be defined as exceedances that can be more obvious or obscure.

*4.3. Bayesian Inference and Simulations for Value at Risk (VaR) and Expected Shortfall (ES)*

Recall that the parameters in the extreme value model are $\theta = (\eta, \sigma, \xi, \mu)$. The prior and posterior distributions are respectively described as follows

4.3.1. The Origin of the Prior Information

Since expressing prior beliefs directly in terms of GPD parameters is a difficult task. The idea to deal with this problem is information within a parameterization on which

experts are familiar. Equation (1) can be re-written as an inversion; thus, we obtain the 1-$p$ quantile of the distribution as follows

$$q = \mu + \frac{\sigma}{\xi}\left(P^{-\xi} - 1\right),\qquad(28)$$

where $q$ is defined as the level of returns associated with a return period of $1/p$ time units. The elicitation of the prior information is expressed in terms of $(q_1, q_2, q_3)$, referring to as the location-scale parameterization of GPD, for specific values of $p_1 > p_2 > p_3$. Consequently, parameters are ordered and $q_1 < q_2 < q_3$. Therefore, the prior information is suggested by setting the median and 90% quantile estimations for specific values of $p$, for example.

Next, the elicited parameters are transformed to gain the equivalent gamma parameters, $d_i \sim Ga(\rho_i, \gamma_i)$ where $i = 1, 2, 3$ and the physical lower bound of the factor is $e_1 = q_0$. $e_1 = 0$ is preferable. The following gamma distributions with hyper parameters: $d_1 = q_1 \sim Ga(\rho_1, \gamma_1)$ and $d_2 = q_2 - q_1 \sim Ga(\rho_2, \gamma_2)$, knowing as the marginal prior distribution for $\sigma$ and $\xi$, which is expressed as follows

$$\begin{aligned}
\pi(\sigma, \xi) &\alpha \left[\mu + \frac{\sigma}{\xi}\left(p_1^{-\xi} - 1\right)\right]^{p_1 - 1} \exp\left[-\gamma_1\left\{\mu + \frac{\sigma}{\xi}\left(p_1^{-\xi} - 1\right)\right\}\right] \\
&\times \left[\frac{\sigma}{\xi}\left(p_2^{-\xi} - p_1^{-\xi}\right)\right]^{p_2 - 1} \exp\left[-\gamma_2\left\{\frac{\sigma}{\xi}\right\}\left(p_2^{-\xi} - p_1^{-\xi}\right)\right] \\
&\times \left[-\frac{\sigma}{\xi^2}\left\{(P_1 P_2)^{-\xi}(\log P_2 - \log P_2) - P_2^{-\xi}\log P_2 + P_1^{-\xi}\log P_1\right\}\right],
\end{aligned}\qquad(29)$$

where $\rho_1, \rho_2, \gamma_1$, and $\gamma_2$ are hyper parameters obtained from the prior information. In the form of the median and some percentiles, the correspondences are the return periods of $\frac{1}{p_1}$ and $\frac{1}{p_2}$. The prior for $q_1$ is in the principle depended on $\mu$. This dependence is substituted by the dependence on the prior mean of $\mu$. Interestingly, in some cases, the situation where $\xi = 0$ is considered. For example, a positive probability is set, and the prior distribution evaluates a probability $q$ if $\xi = 0$ and $1 - q$ if $\xi \neq 0$.

### 4.3.2. The Prior Density for Parameters at the Threshold

Apart from the information above the threshold, $u$ is assigned to follow a truncated normal distribution with parameters $\left(u_\mu, \sigma_\mu^2\right)$, curtailed from below at $e_1$ with density as Equation (30)

$$\pi\left(\mu\,\middle|\,u_\mu, \sigma_\mu^2, e_1\right) = \frac{1}{\sqrt{2\pi\sigma_\mu^2}} \times \left\{\frac{\exp\left[-0.5\left(\mu - u_\mu\right)^2/\sigma_\mu^2\right]}{\Omega\left[-(e_1 - u_\mu)/\sigma_\mu\right]}\right\},\qquad(30)$$

with $u_\mu$ is included in some high percentile and $\sigma_\mu^2$ is sufficient to present a reasonably noninformative prior. In other words, de Zea Bermudez et al. (2001) suggested that the higher level to set the prior distribution for $\mu$, and this requires setting a prior distribution for the hyper thresholds.

### 4.3.3. Posterior Density Estimations

From the expression of the likelihood in Equation (27) and the priors, the posterior distribution is given from using Bayes theorem, which is combined with simulations (the MCMC methods via Metropolis–Hastings algorithms, (Metropolis et al. 1953)). To get hold

of a gamma distribution for data below the threshold, the functional form on the logarithm scale is derived as follows

$$
\begin{aligned}
\log p(\theta|x) = K + \sum_{i=1}^{n} I(x_i < \mu)[\alpha \log \beta - \log \tau(\alpha) + (\alpha - 1) \log x_i - \beta x_i] \\
+ \sum_{i=1}^{n} I(x_i \geq \mu) \log \left( 1 - \int_0^\mu \frac{\beta^\alpha}{\tau(\alpha)} t^{\alpha-1} e^{-\beta t} dt \right) - \sum_{i=1}^{n} I(x_i \geq \mu) \log \sigma \\
- \frac{1+\xi}{\xi} \sum_{i=1}^{n} I(x_i \geq \mu) \log \left[ 1 + \frac{\xi(x_i - \mu)}{\sigma} \right] \\
+ (a-1) \log \alpha - b\alpha + (c-1) \log \frac{\alpha}{\beta} - d\left(\frac{\alpha}{\beta}\right) + \log\left(\frac{\alpha}{\beta^2}\right) \\
- \frac{1}{2}\left(\frac{\mu - u_\mu}{\sigma_\mu}\right) - b_1\left[\mu + \frac{\sigma}{\xi}\left(p_1^{-\xi} - 1\right)\right] \\
+ (a_2 - 1) \log\left[\mu + \frac{\sigma}{\xi}\left(p_2^{-\xi} - p_1^{-\xi}\right)\right] - b_2\left[\mu + \frac{\sigma}{\xi}\left(p_2^{-\xi} - p_1^{-\xi}\right)\right] \\
+ \log\left\{ -\frac{\sigma}{\xi}\left[(p_1 p_2)^{-\xi}(\log p_2 - \log p_1) - p_2^{-\xi} \log p_2 + p_1^{-\xi} \log p_1\right]\right\},
\end{aligned}
\tag{31}
$$

where $k$ is the normalizing constant. For the computation, making analytical posterior distributions depends on the convergence rate in each MCMC simulations case.

### 4.3.4. Risk Measurement

As the goal of the article is the risk analysis for a financial context. The famous Value at Risk (VaR) can summarize the worst loss over a target horizon with a given level of confidence and outline the overall market risk faced by an institution (Assaf 2009).

For extreme data analyses, the GPD continues to boundlessness. This kind of extreme distribution is not known with certainty in practice, but the Bayesian framework allows us to quantify this uncertainty. Expressly, the posterior predictive distribution follows

$$
p\left(x^f \middle| x\right) = \int_\theta p\left(x^f \middle| \theta\right) p(\theta|x) d\theta.
\tag{32}
$$

If uncertainty regarding an unknown parameter is captured in a posterior distribution, a predictive distribution for any quantity $\mu$ that depends on the unknown parameter, through a sampling distribution, can be achieved by the Equation (33). In this case, $p\left(x^f \middle| x\right)$ mentions to an updated GDP observation obtained a set of parameters. The following transformation gives the updated information:

$$
U \sim Uniform(0,1) \geq \log p(\theta|x) = \left[\left(U^{-e} - 1\right) + \frac{\sigma}{e} + \mu\right] \sim GPD(\mu, \sigma, e)
\tag{33}
$$

By the MCMC methods, a large number of large updating samples can be stimulated. In terms of the GDP, the Value at Risk (VaR) and Expected Shortfall (ES) can be the expression as follows:

$$
VaR(1 - \alpha) = \left(\alpha^{-e} - 1\right)\frac{\sigma}{e} + \mu,
\tag{34a}
$$

$$
ES(1 - \alpha) = VaR(1 - \alpha) + \frac{\sigma \alpha^{-e}}{1 - e}.
\tag{34b}
$$

These measures are ordered to obtain quantiles to create intervals. Note that since negative log share returns are included, which are the GPD above a suitable threshold, it is crucial to rescale $\alpha$ by multiplying by the divide between the number of observations and number of exceedances.

## 5. Computational and Comparative Results

### 5.1. Data Visualization

In this section, the historical data tries to explain the type of non-normal distributions. Table 1 details the log-return transformation, which assures the set of observed data

are stationary in long-term periods, and the Phillips-Perron (PP) unit-root test confirms this. Additionally, the expression to define the data set is not normally distributed is represented by the significant level of the Jarque–Bera test. All financial indexes reject the null hypothesis, which refers to the normal distribution.

**Table 1.** Descriptive statistics (original data).

|  | LOG_STI (Singapore) | LOG_SET (Thailand) | LOG_PSEI (Philippines) | LOG_PCI (Indonesia) | LOG_FTSE (Malaysia) |
|---|---|---|---|---|---|
| Mean | 0.001480 | 0.000502 | 0.004399 | 0.010991 | 0.003793 |
| Median | 0.005334 | 0.001854 | 0.011447 | 0.016087 | 0.007192 |
| Maximum | 0.147158 | 0.154828 | 0.147340 | 0.161394 | 0.063923 |
| Minimum | −0.181316 | −0.188425 | −0.166256 | −0.213672 | −0.087201 |
| Std. Dev. | 0.044216 | 0.056605 | 0.048931 | 0.051288 | 0.025348 |
| Skewness | −0.228034 | −0.514831 | −0.455003 | −0.714628 | −1.066256 |
| Kurtosis | 6.995341 | 4.894663 | 4.245375 | 6.527976 | 5.221788 |
| Jarque-Bera | 67.37811 | 19.37479 | 9.912792 | 60.37246 | 39.51644 |
| Probability | 0.000000 | 0.000062 | 0.007038 | 0.000000 | 0.000000 |
| PP-test statistics | −7.761860 | −8.330687 | −8.993564 | −7.475574 | −7.891177 |
| Probability | 0.000000 | 0.000000 | 0.000000 | 0.000000 | 0.000000 |

Source: authors.

### 5.2. The Distribution Outlook Comparison

In this crucial section, adapting from the contribution conducted by Gençay and Selçuk (2004), the threshold is set as 6%, which refers to the approximately understandable return of the stock exchanges. This is the prior information at the threshold line level that explicitly separates exceedance samples and risk-free samples. Shape and scale parameters are estimated from two comparative sources—an original observed distribution (Gaussian random walk) and quantum-wave distribution from each selected financial index. To compute the VaR model at 99% confidence and the corresponding expected shortfall, Table 2 represents the comparative outcome that indicates the modified distribution by applied quantum computing for Bayesian extreme value forecasting can capture missing information more efficiently than the traditional econometrics (Gaussian random walk process) because every DIC values indicate that the quantum distribution of five stock markets in five ASEAN countries is appropriate with the model of Bayesian extreme value prediction compared with data distribution based on the Gaussian random walk process.

**Table 2.** The model validation by Deviance Information Criterion (DIC).

|  |  | Data Distribution Based on Gaussian Random Walk | Data Distribution Based on the Wave Function |
|---|---|---|---|
|  |  | DIC | DIC |
| **Thailand** | **(SET)** | −909.6436 | −934.5987 * |
| **Singapore** | **(STI)** | −910.8061 | −1089.0160 * |
| **Malaysia** | **(FTSE)** | −1288.4080 | −1339.4820 * |
| **Philippines** | **(PSEI)** | −925.4325 | −1032.0220 * |
| **Indonesia** | **(PCI)** | −960.8347 | −1025.6590 * |

Noted: * stands for the minimum value of DIC calculations. Source: authors.

### 5.3. Risk Measures by the Quantum Distribution

In Table 3, it seems to be clear that the risk projections estimated from data sourced by quantum-wave distributions; risk measurements calculated by the VaR model and corresponding expected shortfalls are reported by this table. First, the strong advantage of Bayesian posterior densities is the ability to provide random parametric intervals, which are more suitable for quantile settings in the GPD and random distributions in financial sectors. The 2.5% interval can be applied to stand for the case of risk aversions, the mean

(50%) indicates the risk-neutral case is mentioned, and the 97.5% interval pinpoints the risk lovers. In terms of the investors who need to maximize profits from the markets and protect the minimum risk as much as possible. For the predominant case, investing in Malaysia seems safer than the other four countries, the chance of failures is 16.78% in the case of risk taking, and the opportunity to loss equals 11.37% in risk-avoiding. This is supported by (Pero and Apandi 2018) to introduce Malaysia's leadership role in ASEAN. Malaysia can be deemed as a leader within ASEAN, championing several important policies in the international arena. On the other hand, Thailand's stock exchange is indicated to have the highest rate of losses in both the risk lover case and risk-avoiding case in ASEAN financial markets. The forecasting results are between 16.56% and 30%. Since the financial market partially depends on the situation of business confidence and the political atmosphere. The Thai stock exchange seems to absorb risks more than other ASEAN countries. For Singapore, the Philippines, and Indonesia, risk and return predictions are ranked in the third, fourth, and fifth, respectively. In the scenario of maximizing profit, 24.72% to 26.22% are approximately the taking losses when focusing on the investment in these three markets. Conversely, 14.32% to 15.20% are the chance of losses for the case of risk aversion.

**Table 3.** The extreme value prediction of Value at Risk (VaR) and Expected Shortfall (ES) is based on quantum mechanics.

| | | 2.5% | Mean | 97.5% |
|---|---|---|---|---|
| **Thailand** | VaR at 99% confidence (0.01) | 0.1523 | 0.1940 | 0.2508 |
| | **Expected shortfall** | **(−0.1656)** | **(−0.2201)** | **(−0.3000)** |
| **Singapore** | VaR at 99% confidence (0.01) | 0.1287 | 0.1569 | 0.1982 |
| | **Expected shortfall** | **(−0.1432)** | **(−0.1831)** | **(−0.2472)** |
| **Malaysia** | VaR at 99% confidence (0.01) | 0.1080 | 0.1256 | 0.1496 |
| | **Expected shortfall** | **(−0.1137)** | **(−0.1361)** | **(−0.1678)** |
| **Philippines** | VaR at 99% confidence (0.01) | 0.1399 | 0.1691 | 0.2094 |
| | **Expected shortfall** | **(−0.1544)** | **(−0.1941)** | **(−0.2529)** |
| **Indonesia** | VaR at 99% confidence (0.01) | 0.1377 | 0.1702 | 0.2148 |
| | **Expected shortfall** | **(−0.1520)** | **(−0.1960)** | **(−0.2622)** |

Source: authors.

## 6. Conclusions

Most economic collapses have appeared in many computational predictions relied on raw observed distributions, which are still common sense for traditional econometrics research. The concept of the Gaussian random walk process continues to be suspicious, especially econometrics for stock predictions. In other words, the assumption of distributional normality is sensibly understandable, but it is sensitive to face suspicious predicted outcomes. At the center of the research gap to determine the origin of real data distributions, this article contributes to quantum mechanics applied for matching the wave function, which is relevant to the fundamental processes of thoughts. For this article, risk management in financial analyses is one of the top issues people have struggled to eliminate unquestionably.

Every level of complexities in data science potentially empowers the ability to capture missing information of the quantum-wave distribution. Expressly, the distribution generated by the quantum mechanics done in the wave equation is compatible with Bayesian inference for measuring risks and expected shortfall predictions, especially when exploring for preciseness in ASEAN financial markets. The comparison of DIC strongly secures this statement. With the quantum distribution, it is sensible to state that a realistic parameter is found, a harmonic inference regarding humans' decision making can be computed, and a meticulous estimation for dealing with extreme tails information in raw data can be demonstrated numerically. In conclusion, the quantum distribution can potentially fix the

gap of missing information in data analyses, especially modern econometrics in financial research.

For upcoming research, applying quantum computations in social science is more challenging. The clue that the quantum distribution can give more real observations is the research changer in the age of big-data analyses. The future plan for installing the novel distribution into behavioral economic research and financial econometrics is the major issue that critically confronts traditional aspects.

**Author Contributions:** This contribution is originated conceptually by C.C. In other words, C.C. has done major parts of the paper, including investigation, methodology, software, supervision, and project administration. He is also the corresponding author. For the sections regarding data curation, writing–original draft preparation, review and editing, and visualization, S.W. takes charge of these tasks. He is the co-author. Both authors have read and agreed to the published version of the manuscript.

**Funding:** This research received no external funding.

**Data Availability Statement:** Data in finance, business and economics.

**Acknowledgments:** This research was supported by a research grant from Chiang Mai University.

**Conflicts of Interest:** The authors declare no conflict of interest.

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
