# Peer review of "Applying Quantum Mechanics for Extreme Value Prediction of VaR and ES in the ASEAN Stock Exchange†"

_economies, doi:10.3390/economies9010013_

Round 1

Reviewer 1 Report

Dear Authors,

This manuscript purpose of this article is to devise methodically a quantum-wave distribution to better analyse risks 9 and returns for stock markets in ASEAN countries, including Thailand (SET), Singapore (STI), 10 Malaysia (FTSE), Philippines (PSEI), and Indonesia (PCI).

  1. The Tangible Existence of Econophysics and Quantum Mechanics

I propose to the authors the renaming of section 1 as ,, Introduction ,,. This section including an analytical framework for the research.

  1. Literature and Critical thinking

The authors also distinguish the contribution to the existing literature and the gap.  However, the authors did not specify the source figure 4. I recommend a more detailed explanation of figure 4 (line 126).

  1. The Objective and Scope of Research

The authors specify the variables for use in the study, with supported definitions and measurements. However, the authors did not specify the source of figure 6 (line 177). I recommend a more detailed explanation of figure 6.

   4.Methodology

The hypotheses and methods are well described. The authors present a good methodological approach. However, the authors did not specify the source of figure 7 (line 249).

   5.Computational and Comparative Results

The authors specify the result for use in the study.  I recommend a more detailed explanation of Table 3 (line 396).

  1. Exclusive Summary

I propose to the authors the renaming of section 6 as ,,Conclusions,,

The authors present the results of the analyses with good explanation and description. This section provides a good overview of the research findings, with an adequate assessment of the findings within the contexts of the existing literature.  I recommend:

-including a new paragraph on discussions about future implications and research;

-if there are significant methodological limitations in data collection and analysis, they should be mentioned and addressed here.

Best regards,

Reviewer

Author Response

Dear Reviewer

I have already made a change to some information in my research article follow your comments, sir. Please recheck again from the attached file

Your comments are very useful for the fulfillment of my research article.

Thank you so much.

Sincerely.

Reviewer 2 Report

Please find attached the comments.

Author Response

(The authors gave the same response as above.)
